# Anethole inhibits human U87 Glioma cell proliferation by inducing apoptosis via the PI3K/AKT pathway

**Ahmed Abdullah Al Awadh**[1], **Elhashimi Eltayb Hassan**[1], **Omer Mohamed Shoaib**[1],
**Osman A.E. Elnoubi**[1], **Saadalnour Abusail Mustafa**[1], **Yasir Mohammed Althayrayan**[2,3],
**Majed Ahmed Althayrayan**[4], **Mohammed Merae Alshahrani**[1]*

1 Department of Clinical Laboratory Sciences, Faculty of Applied Medical Sciences, Najran University, Najran, Saudi Arabia, 2 Director of Administration, College of Applied Medical Sciences, Najran University, Najran, Saudi Arabia, 3 Laboratory Technician, Department of Clinical Laboratory Sciences, Najran University, Najran, Saudi Arabia, 4 Laboratory Technician, University Hospital, Najran University, Najran, Saudi Arabia

* mmalshahrani@nu.edu.sa

## Abstract

Glioma is characterized by rapid progression, resistance to conventional therapies, and poor prognosis. Current treatments are often limited by their inability to selectively target tumor cells. Natural compounds, such as anethole, have shown promising anticancer properties in various cancers, but their efficacy in glioblastoma remains unexplored. This study investigates the anticancer activity of anethole in glioma cells, focusing on its influence on cell proliferation, apoptosis, and the PI3K/Akt pathway. Human glioma cell lines (U87-MG and LN-229) and normal human astrocytes (NHA) were treated with anethole. Cell viability was evaluated using the CCK-8 assay, while colony formation and AO/EB staining assays evaluated proliferation and apoptosis, respectively. Cell viability was evaluated using the CCK-8 assay, while colony formation and AO/EB staining assessed proliferation and apoptosis, respectively. Western blotting was used to analyze apoptosis-related markers and PI3K/AKT pathway proteins. Molecular docking assessed anethole–PI3K binding, and in silico analyses (SwissTargetPrediction, KEGG, RummaGEO) identified putative targets and pathways. Anethole exhibited selective cytotoxicity, with significantly lower $IC_{50}$ values for U87-MG (10.8±0.42 µM) and LN-229 (12.5±0.51 µM) compared to NHA cells (61.5±1.27 µM), calculated from three independent experiments with triplicate wells. Colony formation was notably inhibited in a dose-dependent manner. AO/EB staining and Western blotting confirmed this with upregulation of Bax, downregulation of Bcl-2, and reduced phosphorylation of PI3K and Akt. Molecular docking revealed strong binding affinity of anethole to PI3K (−9.32 kcal/mol), and Western blot showed inhibition of PI3K and Akt phosphorylation. Anethole selectively inhibits glioma cell proliferation by inducing apoptosis and suppressing the PI3K/Akt cascade. These

**Data availability statement:** All relevant data are within the manuscript.

**Funding:** This work was supported by the Deanship of Graduate Studies and Scientific Research at Najran University under the Easy Funding Program (grant code NU/EFP/MRC/13/220) awarded to A.A.A. The funder had no role in study design, data collection and analysis, decision to publish, or preparation of the manuscript.

**Competing interests:** The authors have declared that no competing interests exist.

observations underscore its potential as a novel therapeutic agent for glioblastoma, warranting further preclinical and clinical investigations.

## 1. Introduction

Gliomas are among the most prevalent primary brain tumors, constituting about 80% of all malignant intracranial neoplasms [1]. Glioblastomas, a subtype of gliomas, are particularly malignant, characterized by excessive proliferation, invasion into adjacent brain parenchyma, and resistance to standard therapies, leading to a grim prognosis and a 5-year survival rate of only 2–10% despite the best possible treatment. The prevalence of gliomas globally is increasing, raising the critical need for new therapeutic approaches to combat this devastating illness [2]. Given the limited efficacy of current treatment modalities, including surgery, radiotherapy, and chemotherapy, there is a critical demand for new agents that can effectively target glioma cells while minimizing toxicity to normal brain tissue [2]. Conventional chemotherapeutics such as temozolomide also have limitations, including drug resistance and side effects [3]. These shortcomings emphasize the need for new therapeutic strategies that overcome these gaps and enhance patient outcomes. In the quest for novel treatments, natural products are recognized as a promising reservoir of bioactive molecules with potent anticancer properties [4]. Plant-derived compounds, in particular, have shown remarkable antitumor activity by inducing multiple mechanisms of action such as apoptosis, cell-cycle arrest, and signaling pathway modulation [5]. The effectiveness of compounds like paclitaxel and vincristine illustrates the value of natural molecules as important resources for cancer drug development [6].

Anethole, a plant-derived phytoconstituent with a broad range of therapeutic applications [7], has recently emerged as a topic of great interest. Several studies have highlighted its ability to target multiple cellular pathways involved in cancer progression, making it a strong candidate for further exploration in oncology. Recent studies have shown the potency of anethole against several types of cancers, such as prostate and oral cancers [8,9]. For example, Contant et al. (2021) showed that anethole activates apoptosis in oral cancer cells, while Nakagawa and Suzuki (2003) reported its ability to impede growth via cell-cycle arrest in prostate cancer cells [8,9]. Furthermore, anethole has also been reported to activate apoptosis in breast cancer cells and inhibit the metastasis of prostate cancer cells [10,11]. In yet another study, anethole has been shown to target STAT3 to impede lung cancer cell growth [12]. These findings underscore the versatility of anethole as an anticancer molecule**.** However, despite its reported efficacy in various cancers, its effects on gliomas remain unexplored. This lack of data highlights a critical gap in understanding the potential of anethole in brain tumors. Among the numerous signaling pathways implicated in glioblastoma pathogenesis, the phosphoinositide 3-kinase (PI3K)/AKT pathway is one of the most frequently activated [13]. This pathway regulates key cellular processes such as proliferation, apoptosis, angiogenesis, metabolism, and therapy resistance [14]. Genetic alterations, such as PTEN loss and PIK3CA amplification, contribute to its hyperactivation in gliomas [15]. Importantly, PI3K/AKT signaling has

been associated with tumor aggressiveness, treatment failure, and poor prognosis in glioblastoma patients. Therefore, the PI3K/AKT axis is considered a prime molecular target for therapeutic intervention in glioblastoma. Interfering with this pathway may sensitize glioma cells to apoptosis and reduce their invasive capacity [16].

Accordingly, the aim of this investigation was to evaluate the anticancer efficacy of anethole against glioma cells and to determine its potential mechanism of action. Specifically, this research aimed to unveil the influence of anethole on cell growth and key molecular cascades, such as the PI3K/AKT pathway, which plays a central role in glioma pathogenesis [17]. By targeting this pathway, we sought to determine whether anethole could overcome resistance mechanisms and lay the foundation for its future use as a therapeutic agent in glioma therapy. In doing so, the present study contributes to the expanding literature on natural product-based approaches to cancer therapy and may pave the way for future clinical investigations.

## 2. Materials and methods

### 2.1. Drug and reagents

Anethole (99% purity; Sigma-Aldrich, Oakville, ON, Canada) was dissolved in methanol to prepare a 3 mM stock solution and subsequently diluted to the required concentrations for each experiment. U87-MG (ATCC® HTB-14™), LN-229 (ATCC® CRL-2611™), and normal human astrocytes (NHA; ScienCell Cat.#1800) were authenticated and confirmed mycoplasma-free before use. DMEM (Gibco), FBS (Gibco), penicillin-streptomycin (Gibco), CCK-8 (Dojindo Cat.#CK04), crystal violet (Sigma), AO/EB (Sigma), RIPA buffer (Thermo), PVDF membranes (Millipore), and ECL substrate (Thermo) were used for all assays.

### 2.1. Cell culture

Glioma cell lines U87-MG and LN-229, along with NHA, were cultured in DMEM supplemented with 10% FBS and 1% penicillin–streptomycin at 37 °C in a humidified 5% $CO_2$ incubator. Prior to anethole treatment, cells were seeded and grown to 70–80% confluence.

### 2.2. CCK-8 assay

Cell viability was determined using the CCK-8 assay. U87-MG, LN-229, and NHA cells were seeded in 96-well plates ($5 \times 10^3$ cells/well) and treated with anethole (0–96 µM) for 24 h [8]. After treatment, 10 µL of CCK-8 solution was added and plates were incubated for 2 h at 37 °C. Absorbance at 450 nm was measured using a microplate reader. Each experiment contained three technical replicate wells per condition; values were averaged to yield one biological replicate.

### 2.3. Colony assay

U87-MG cells were plated in six-well plates with 500 cells per well for colony development analysis. Cells were exposed to anethole at concentrations of 0, 5.4, 10.8, and 21.6 µM and cultured for 10–14 days, with the medium refreshed every 2–3 days. Colonies were fixed with 4% paraformaldehyde for 20 minutes, stained with 0.5% crystal violet, and observed by a microscope. Those with over 50 cells were manually counted. The assay was repeated n = 3 independent times, each with two technical wells per condition. Colony counts were averaged per experiment before statistical analysis.

### 2.4. AO/EB staining assay

To determine the induction of apoptosis, AO/EB staining was performed. U87-MG cells were administrated with anethole at 0, 5.4, 10.8, and 21.6 µM for 24 h. The treated cells were washed with PBS and stained with AO/EB solution (1:1 ratio) for 6 min. The stained cells were visualized under a fluorescence microscope, and apoptotic, necrotic, and viable cells were quantified based on fluorescence emission.

Apoptotic, necrotic, and viable cells were differentiated based on nuclear morphology and fluorescence emission: viable cells appeared green with uniform nuclei, early apoptotic cells displayed green nuclei with chromatin condensation, late apoptotic cells showed orange-red fluorescence with nuclear fragmentation, and necrotic cells exhibited uniform orange-red nuclei. At least 300 cells per group were counted in randomly selected fields using ImageJ [18]. The percentage reported represents total apoptosis (early + late) averaged from three independent experiments.

## 2.5. *In Silico* analysis

Molecular targets of anethole were identified using the SwissTargetPrediction platform. Identified targets were subjected to KEGG pathway enrichment analysis to elucidate pathways potentially influenced by anethole. Additionally, gene expression profiles were examined using the RummaGEO database to correlate identified targets with cancer related processes. The expression levels of the selected anethole targets were retrieved from the TCGA database. The pharmacokinetic properties of anethole were evaluated using the SwissADME web tool (http://www.swissadme.ch). The SMILES structure of anethole was entered to obtain predictions on lipophilicity (LogP), topological polar surface area (TPSA), gastrointestinal (GI) absorption, and blood–brain barrier (BBB) permeability. A consensus LogP was calculated from five models, and BBB permeability was assessed using the BOILED-Egg model. Additional parameters such as water solubility, CYP enzyme inhibition, and bioavailability score were also recorded.

## 2.6. Molecular docking

Docking simulations were performed using DockingServer. Anethole was energy-minimized with the MMFF94 force field, and Gasteiger charges were assigned. The protein model was prepared in AutoDock tools with hydrogen atoms and Kollman charges [19,20]. A 20 × 20 Å grid with 0.375 Å spacing was used for affinity maps. Simulations employed the Lamarckian genetic algorithm and Solis & Wets local search method, with random ligand initialization, two runs, 250,000 energy evaluations, and a population size of 150 [20].

AutoDock 4.2 was chosen over more recent tools like AutoDock Vina or Schrödinger due to its well-established reliability, flexibility in algorithm customization, and detailed output of binding energy components. While Vina offers faster performance, AutoDock 4.2 provides more granular control over docking parameters and energy scoring, which was essential for comparative analysis of binding interactions. Moreover, web-based interface of DockingServer ensures reproducibility and accessibility, making it suitable for high-throughput screening in academic settings.

## 2.7. Western blot analysis

Western blotting was employed to examine expression of specific proteins. Cells were lysed using RIPA buffer and 30 µg of protein was loaded onto an SDS-PAGE gel, transferred to PVDF membranes, and blocked with 5% non-fat milk. Membranes were incubated overnight at 4°C with primary antibodies targeting Bax (Cell Signaling #2772), Bcl-2 (#15071), PI3K p85 (#4257), p-PI3K (Tyr458) (#4228), AKT (#9272), p-AKT (Ser473) (#9271), and β-actin (#4967). After washing, HRP-conjugated secondary antibodies were applied, and protein bands were visualized using ECL. Densitometric analysis was performed using ImageJ. Phospho-proteins were normalized to their respective total proteins (p-PI3K/PI3K and p-AKT/AKT), with β-actin serving only as a loading control. Uncropped blots with visible molecular weight markers were included in (Supplementary S1 Fig).

## 2.8. Statistical analysis

All statistical analyses were performed using GraphPad Prism software version 9.5.1 (GraphPad Software, LLC, San Diego, CA, USA). Data are presented as mean ± SD of n = 3 independent biological experiments, unless otherwise stated. Comparisons between groups were conducted using either Student's t-test or one-way ANOVA, followed by Tukey's post hoc test where appropriate. A p-value less than 0.05 was considered statistically significant.

## 3. Results

### 3.1. Anethole inhibits proliferation of glioma cells

The antiproliferative effects of anethole were assessed on U87-MG, LN-229, and normal NHA cells. Anethole treatment for 24 hours significantly decreased glioma cell viability in a concentration-dependently ($p < 0.05$). The $IC_{50}$ values, calculated from three independent experiments with triplicate wells (n = 3), were 10.8 ± 0.42 µM for U87-MG cells, 12.5 ± 0.51 µM for LN-229 cells, and 61.5 ± 1.27 µM for NHA cells (Fig 1A). This highlights the selective cytotoxicity of anethole toward glioma cells compared to normal astrocytes. The colony formation assay further confirmed these results, showing a significant reduction in colony numbers in U87-MG cells treated with 5.4, 10.8, and 21.6 µM anethole compared to untreated controls (Fig 1B).

### 3.2. Anethole triggers apoptosis in glioma cells

The pro-apoptotic effects of anethole were confirmed through AO/EB staining to visually quantify apoptotic cells. AO/EB staining showed a dose-dependent increase in apoptotic cells, with higher concentrations of anethole resulting in more

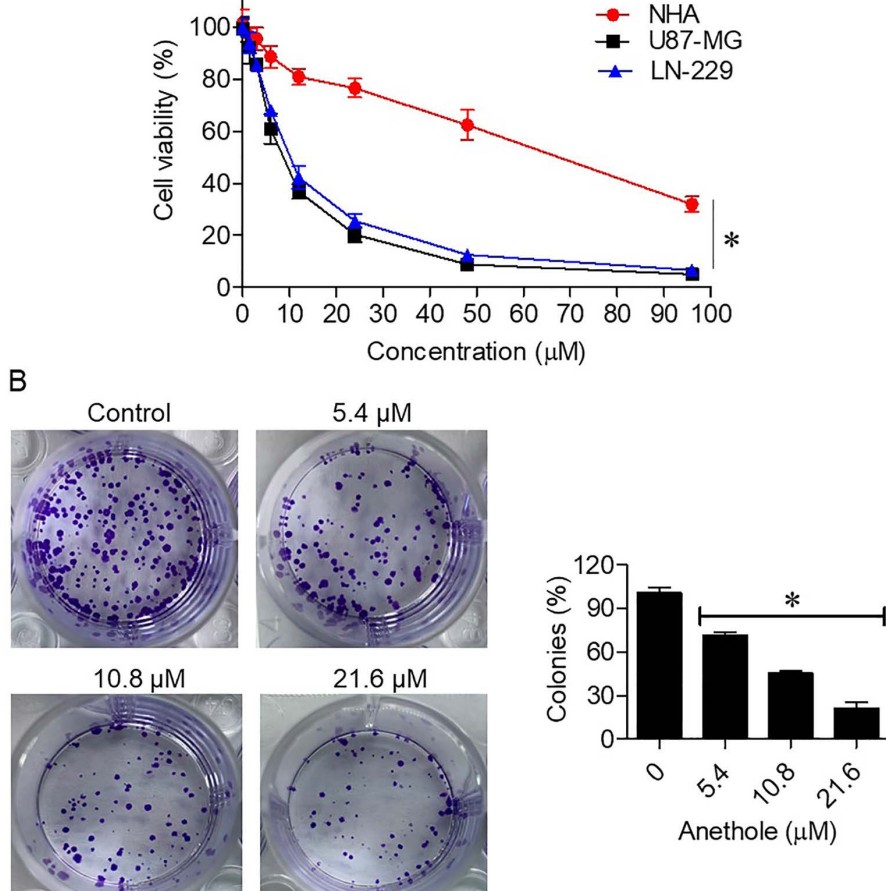

**Fig 1. Anethole inhibits the proliferation of glioma cells. (A)** Cell viability assay showing the effects of anethole on glioma (U87-MG, LN-229) and normal (NHA) cells. **(B)** Colony formation assay of U87-MG cells treated with indicated concentrations of anethole. Each experiment was independently repeated three times, with three technical replicates per condition. Data are expressed as the mean ± SD of three independent experiments (*P < 0.05 vs. control).

yellow-orange and red fluorescent cells, indicating apoptosis in U87-MG cells. The percentage of total apoptotic cells (early+late) was~2% at control which increased to 51.3% at 21.8 μM anethole (Fig 2A). Western blot analysis corroborated these findings, revealing a dose-dependent Bax upregulation and Bcl-2 downregulation (Fig 2B-2D).

### 3.3. Anethole targets genes involved in cancer pathways

Through SwissTargetPrediction, 98 potential targets of anethole were identified, with functional roles in enzymatic activity, gene regulation, and redox processes (Fig 3A). KEGG pathway enrichment analysis showed that these targets are associated with cancer-related pathways, including chemical carcinogenesis, xenobiotic metabolism, and immune checkpoint regulation, emphasizing their roles in apoptosis and cancer immunotherapy (Fig 3B). RummaGEO analysis further demonstrated that these targets are significantly expressed in glioma tissues and are linked to key processes such as apoptosis, autophagy, cell cycle regulation, EMT, and metastasis, highlighting their relevance in glioma progression and treatment (Fig 3C). KEGG cluster analysis revealed 16 of these genes to be involved in cancer pathways (Fig 4A).

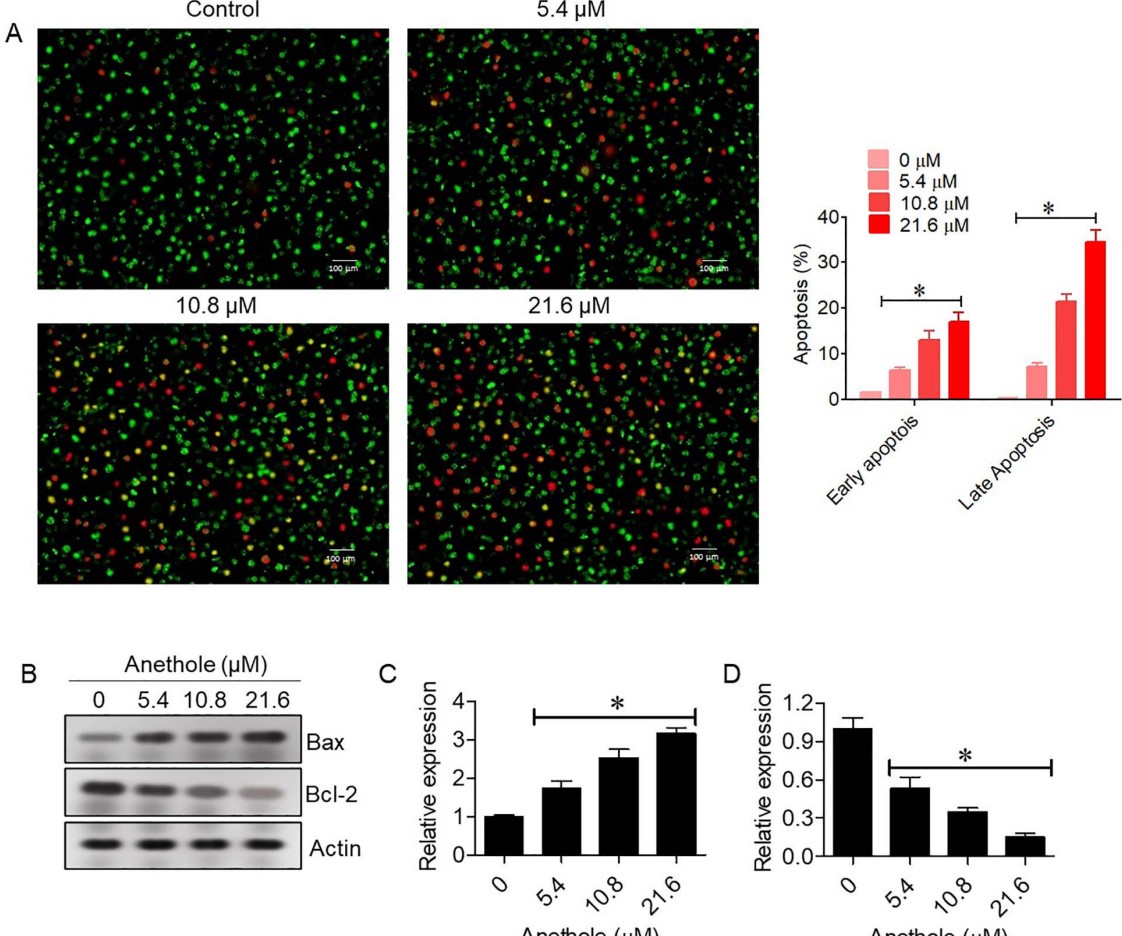

**Fig 2. Anethole induces apoptosis in glioma cells. (A)** AO/EB staining assay showing apoptosis in U87-MG cells at indicated concentrations of anethole. **(B)** Western blot analysis of Bax and Bcl-2 protein expression in U87-MG cells treated with indicated concentrations of anethole. **(C)** Densitometric analysis of Bax expression. **(D)** Densitometric analysis of Bcl-2 expression. All experiments were independently repeated three times, and data are presented as mean±SD (*P<0.05 vs. control).

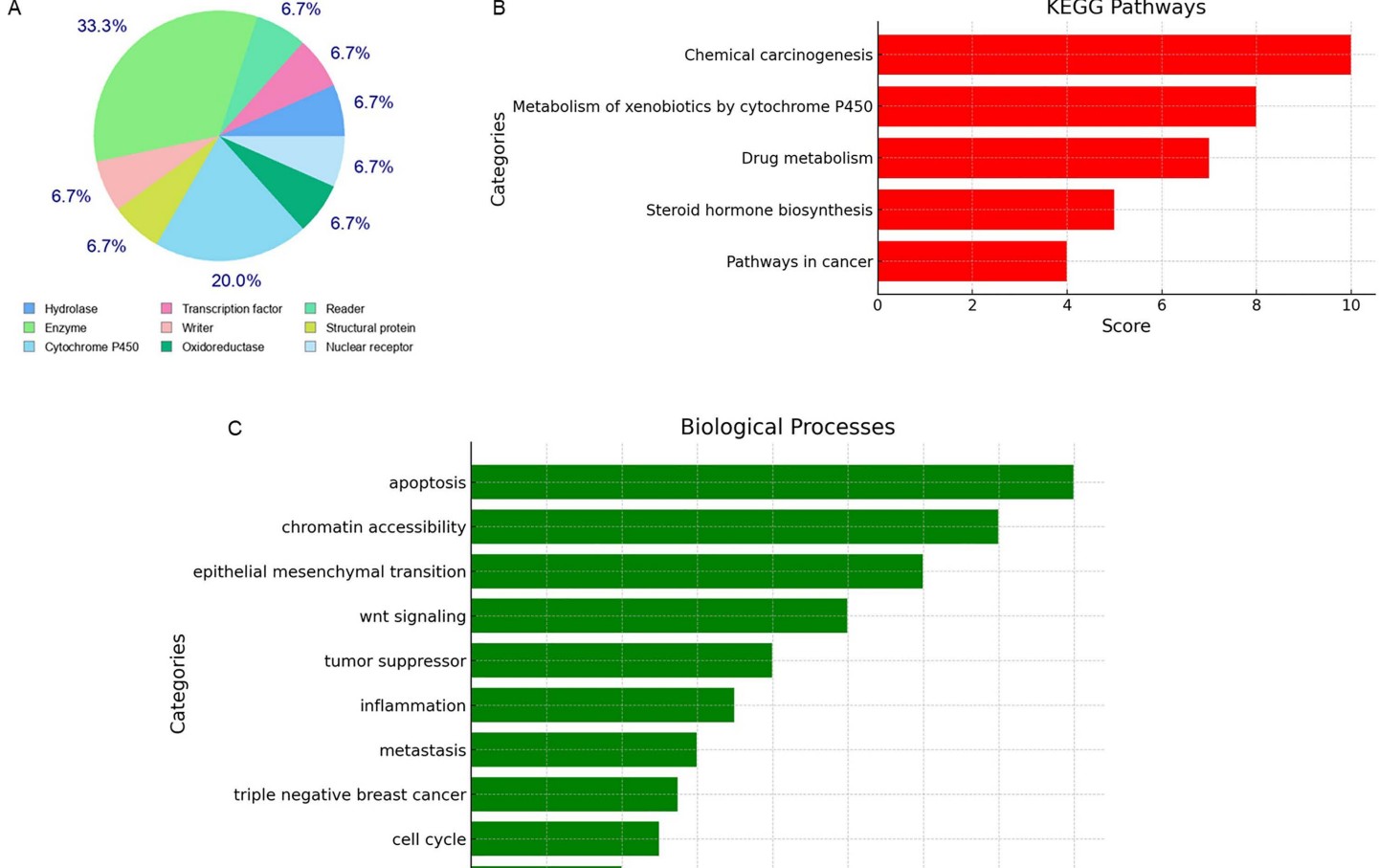

**Fig 3. Target identification of anethole. (A)** Pie chart representing the percentage distribution of targets identified by the SwissTarget Prediction tool across different categories. **(B)** KEGG pathway analysis showing the association of identified targets with various pathways. **(C)** RummaGEO analysis highlighting the relevance of identified targets to molecular processes.

Additionally, TCGA data analysis confirmed differential expression patterns of these targets in glioma, with key oncogenic pathways such as PI3K/Akt, JAK/STAT, and hormone biosynthesis being significantly enriched (Fig 4B). These integrated findings corroborate the experimental results of apoptosis induction by anethole, underscoring its potential as an anticancer agent targeting glioma-specific pathways.

### 3.4. Anethole targets PI3K and inhibits the PI3K/Akt pathway

Of the targets identified, PI3K and JAK2 were prioritized based on their elevated expression in glioma datasets from TCGA. Anethole demonstrated differential binding affinities with PI3K and JAK2 in molecular docking studies, highlighting its potential as a modulator of key signaling pathways. For PI3K, anethole exhibited a strong binding interaction with an estimated free energy of binding of −9.32 kcal/mol and an inhibition constant (Ki) of 120.40 nM. The binding was stabilized primarily by van der Waals, hydrogen bonding, and desolvation energies (−10.15 kcal/mol), with minor electrostatic

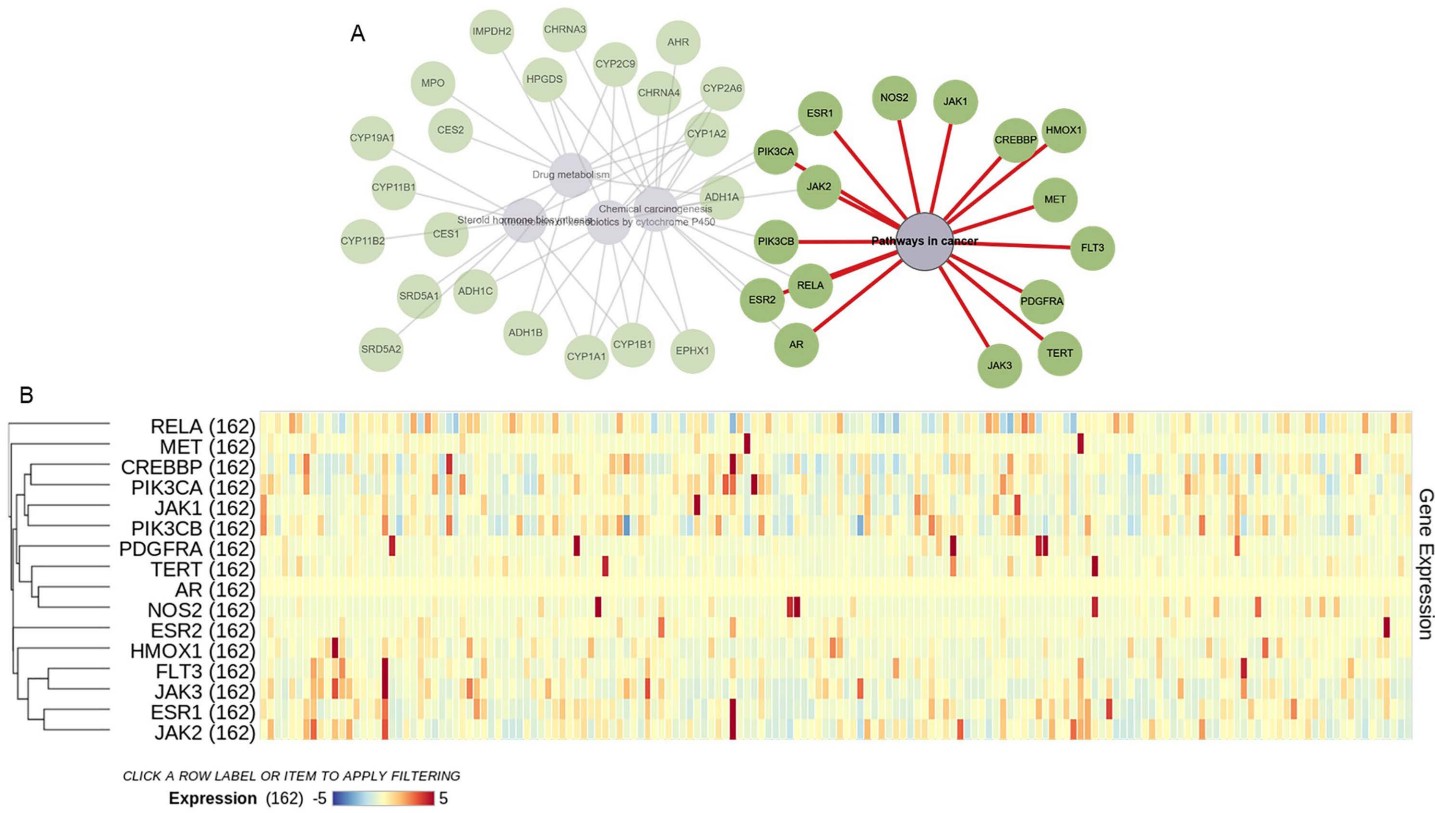

**Fig 4. Identification and expression of cancer-related targets of anethole. (A)** KEGG cluster analysis illustrating 16 cancer-related targets of anethole. **(B)** Expression levels of these 16 targets in 162 glioma tissues retrieved from the TCGA database.

contributions (−0.50 kcal/mol), resulting in a total intermolecular energy of −10.65 kcal/mol. The interaction surface was 650.471, and key residues involved were TRP780, ILE800, TYR836, ILE848, VAL850, VAL851, SER854, MET922, and ILE932, emphasizing significant contact with the active site (Fig 5A). In contrast, the interaction with JAK2 was moderate, with a free energy of binding of −4.03 kcal/mol and a Ki of 1.11 mM. The binding energies were mainly derived from van der Waals and hydrogen bonding (−4.36 kcal/mol) with minor electrostatic interactions (−0.26 kcal/mol), leading to a total intermolecular energy of −4.62 kcal/mol. The interaction surface was smaller (539.088), with key residues including ARG975, LEU1001, PRO1002, LYS1005, and PHE1031 (Fig 5B). These findings suggest that anethole has a stronger binding affinity and more favorable interactions with PI3K compared to JAK2, supporting its potential as a selective inhibitor of PI3K-related pathways while exhibiting moderate activity against JAK2. Further optimization could enhance its efficacy and specificity for these targets. Western blot analysis further confirmed the inhibitory influence of anethole on the PI3K/Akt pathway (Fig 5C). Densitometric analysis, performed by normalizing phosphorylated proteins to their respective totals (p-PI3K/PI3K and p-AKT/AKT) with β-actin serving as a loading control, revealed that anethole treatment induced a concentration-dependent decrease in PI3K and AKT phosphorylation in U87-MG cells, indicating effective suppression of this critical survival pathway (Fig 5D and 5E).

### 3.5. *In Silico* ADME and BBB Prediction

SwissADME analysis revealed that anethole has a consensus LogP of 2.79, indicating favorable lipophilicity for membrane permeability. The topological polar surface area (TPSA) was calculated as 9.23 Å², well below the 90 Å² threshold

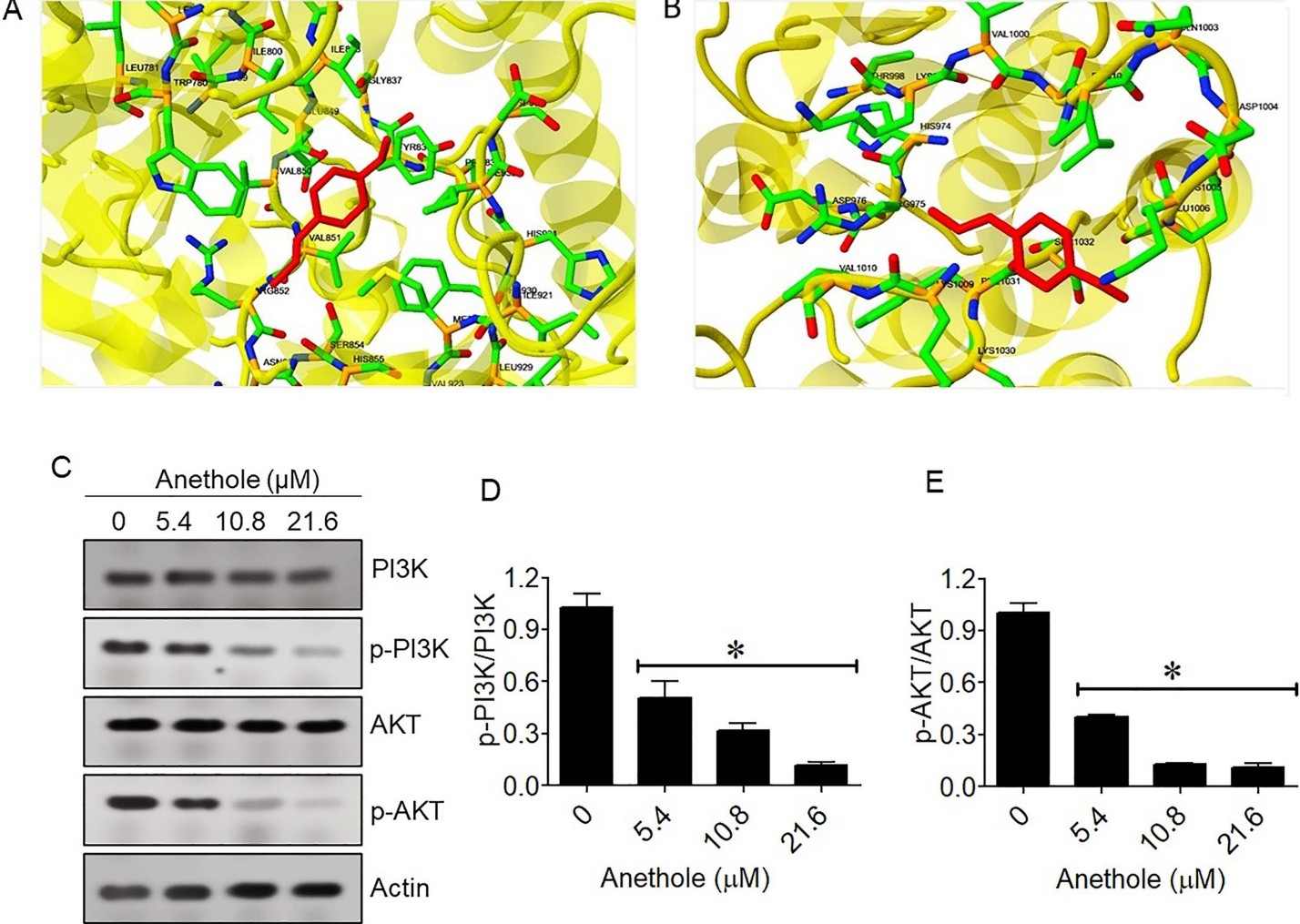

**Fig 5. Anethole targets the PI3K/AKT pathway in glioma cells. (A)** Molecular docking analysis showing the interaction of anethole with PI3K. **(B)** Molecular docking analysis showing the interaction of anethole with JAK2. **(C)** Western blot analysis of PI3K/AKT pathway-related proteins in U87-MG cells treated with anethole. **(D)** Densitometric quantification of p-PI3K/PI3K. **(E)** Densitometric quantification of p-AKT/AKT. Data are expressed as mean ± SD (n = 3 independent biological experiments). Statistical significance was determined by one-way ANOVA with Tukey's post hoc test (*P < 0.05).

for effective blood–brain barrier (BBB) penetration (Supplementary S2 Fig in S1 Fig). The BOILED-Egg model predicted that anethole is BBB permeant and exhibits high gastrointestinal absorption (Supplementary Fig S3 in S1 Fig). Additionally, the compound was found to be soluble across all three solubility models and showed a bioavailability score of 0.55. These findings support the potential of anethole to cross the BBB and exert central nervous system effects.

## 4. Discussion

Glioma continues to be a major oncology challenge due to its aggressive growth, therapy resistance, and dismal prognosis [21]. Natural compounds such as anethole have gained attention for their potential to modulate critical cancer pathways with minimal toxicity to normal cells [22]. The present study emphasizes the strong anticancer activity of anethole, its selective cytotoxicity, apoptosis induction, and the inhibition of the PI3K/Akt cascade in glioma cells. These results

establish a basis for understanding the molecular mechanisms underlying the therapeutic potential of anethole and its broader applications in glioma therapy. In this study, anethole exhibited preferential cytotoxicity toward glioma cells as indicated by significantly lower $IC_{50}$ values for U87-MG and LN-229 compared with normal human astrocytes (NHA). This apparent selectivity may be attributed to the higher basal activation of the PI3K/AKT pathway and altered membrane permeability in glioma cells. Tumor cells often depend on constitutively active PI3K/AKT signaling due to genetic alterations such as PTEN loss or PIK3CA amplification, rendering them more vulnerable to inhibitors targeting this axis [23,24]. Additionally, changes in lipid composition, increased fluidity, and disrupted oxidative homeostasis in cancer membranes may facilitate greater anethole uptake compared with normal astrocytes [25]. These observations are consistent with earlier studies showing selective cytotoxicity of anethole in oral and breast cancer models [8,26]. The decreased colony formation of U87-MG cells further substantiates its antiproliferative action, in agreement with Elkady (2018), who demonstrated similar effects in prostate cancer cells [27].

Apoptosis induction, a hallmark of anticancer efficacy [28], was validated in the current study through AO/EB staining, revealing a dose-dependent increase in apoptotic cells with typical nuclear condensation and fragmentation. Western blot analysis showed elevated Bax and reduced Bcl-2 expression, indicating activation of intrinsic apoptosis. These findings align with prior reports demonstrating modulation of the Bax/Bcl-2 ratio by anethole [29] and its ability to regulate PTEN and CXCR4 to trigger apoptosis [11].

Unlike earlier suggestions of a pro-oxidant role [27], accumulating evidence indicates that anethole predominantly exhibits antioxidant activity, decreasing reactive oxygen species (ROS) levels and enhancing glutathione (GSH) defenses in cancer cells [8]. These antioxidant effects may indirectly contribute to apoptosis by rebalancing redox signaling and suppressing pro-survival pathways rather than by inducing oxidative stress [30,31]. Hence, the apoptotic mechanism observed in glioma cells in this study is more consistent with direct inhibition of the PI3K/AKT cascade and modulation of apoptotic regulators, rather than ROS elevation.

In the current work, in silico analysis identified 98 potential molecular targets of anethole, with KEGG pathway enrichment revealing 16 cancer-associated targets. This bioinformatic evidence strengthens the experimental findings and suggests that anethole exerts multi-targeted effects beyond a single signaling node. The RummaGEO analysis further linked these targets to apoptosis, cell cycle control, and autophagy, supporting the observed biological outcomes.

The PI3K/Akt cascade, frequently dysregulated in gliomas, is pivotal in regulating survival and proliferation [16]. Molecular docking demonstrated a strong binding affinity of anethole to PI3K with favorable interaction energy, supporting its potential as a direct inhibitor. Western blot analysis confirmed a dose-dependent reduction in PI3K and AKT phosphorylation, consistent with the in silico findings. These results align with previous reports by Ha et al. (2014) and Arumugam et al. (2021), who found that anethole suppressed the PI3K/Akt pathway in prostate and breast cancer cells, respectively [10,32]. Through this pathway, anethole may sensitize glioma cells to apoptosis and enhance the efficacy of combination regimens with standard chemotherapeutics. When compared to established PI3K inhibitors such as Wortmannin and LY294002, anethole displayed moderate but meaningful binding affinity with a lower likelihood of off-target cytotoxicity. These synthetic inhibitors, while potent, are limited by instability, poor solubility, and systemic toxicity. The natural origin, favorable pharmacokinetic properties, and selectivity profile of anethole may provide a therapeutic advantage, supporting its potential as a safer alternative or adjunct agent in glioma management [33–36].

The findings of this study hold important clinical relevance for the development of anethole-based therapies against gliomas. The selective cytotoxicity of anethole toward glioma cells, coupled with its ability to modulate multiple oncogenic signaling pathways, underscores its promise as a novel therapeutic candidate. Moreover, its predicted lipophilicity and potential to traverse the blood–brain barrier (BBB) suggest suitability for treating central nervous system (CNS) tumors [37]. Although previous pharmacokinetic data support possible CNS access for anethole derivatives [37,38], dedicated permeability studies—such as in vitro BBB models and *in vivo* brain-to-plasma distribution assays—are required to confirm its bioavailability within the brain. Despite favorable computational predictions, clinical translation may be constrained

by anethole's limited aqueous solubility and extensive first-pass metabolism following oral administration. To overcome these barriers, advanced delivery strategies, including lipid-based nanoformulations, prodrug development, or intranasal delivery systems, could enhance its systemic bioavailability and CNS penetration [39,40]. Future pharmacokinetic investigations should focus on validating these approaches to optimize the therapeutic potential of anethole for glioma management.

Collectively, the findings of this study highlight anethole as a promising candidate for glioma therapy. However, certain limitations should be acknowledged. First, the current work did not include apoptosis assays in NHA, which would have provided additional confirmation of cancer-specific selectivity. Second, we did not employ quantitative flow cytometry (Annexin V/PI) to distinguish early and late apoptotic events, and we recognize this as a methodological limitation. Third, while the inhibition of PI3K/AKT phosphorylation supports the proposed mechanism, we did not perform a rescue experiment (e.g., constitutively active AKT transfection) to definitively prove pathway causality. These aspects are important for future validation, along with *in vivo* efficacy studies, real-time binding assays (ITC, SPR, or BLI), and investigation of immune and angiogenic responses within the tumor microenvironment.

## 5. Conclusion

In conclusion, this study provides compelling evidence that anethole suppresses glioma cell proliferation and induces apoptosis by inhibiting the PI3K/AKT signaling pathway. Its selective cytotoxicity toward glioma cells, combined with favorable in silico pharmacokinetic predictions, supports its potential as a candidate for glioma therapy. While *in vivo* and mechanistic rescue experiments remain to be performed, the present findings establish a robust foundation for further translational and preclinical evaluation. Future studies should focus on optimizing drug delivery across the BBB and exploring synergistic effects of anethole with established chemotherapeutic agents to develop safer, more effective strategies against glioblastoma and other CNS malignancies.

**Institutional review board approval and informed consent**

This study did not involve human participants, patient data, or clinical samples. Only commercially available human glioma cell lines (U87-MG and LN-229) and normal human astrocytes (NHA) were used. Therefore, Institutional Review Board (IRB) approval and informed consent were not required.

**Supporting information**

**S1 Fig. Uncropped blots with visible molecular weight markers.**
(DOCX)

**S2 File. *SwissADME analysis of anethole.*** Predicted physicochemical, pharmacokinetic, and drug-likeness properties of anethole were evaluated using the SwissADME web tool based on its SMILES structure.
(DOCX)

**S3 File. *BOILED-Egg plot prediction o gastrointestinal absorption and brain penetration of anethole.*** The plot shows anethole's predicted position within the yellow region, indicating blood–brain barrier (BBB) permeability. The white region represents high gastrointestinal absorption (HIA). The red circle (PGP–) marks anethole as a non-substrate of P-glycoprotein.
(DOCX)

**Acknowledgments**

The authors are thankful to the Deanship of Graduate Studies and Scientific Research at Najran University for funding this work.

## Author contributions

**Conceptualization:** Ahmed Abdullah Al Awadh, Mohammed Merae Alshahrani.

**Data curation:** Ahmed Abdullah Al Awadh, Elhashimi Eltayb Hassan, Omer Mohamed Shoaib, Osman A.E. Elnoubi, Mohammed Merae Alshahrani.

**Formal analysis:** Ahmed Abdullah Al Awadh, Elhashimi Eltayb Hassan, Omer Mohamed Shoaib, Osman A.E. Elnoubi, Saadalnour Abusail Mustafa, Yasir Mohammed Althayrayan, Majed Ahmed Althayrayan, Mohammed Merae Alshahrani.

**Funding acquisition:** Ahmed Abdullah Al Awadh, Mohammed Merae Alshahrani.

**Investigation:** Ahmed Abdullah Al Awadh, Mohammed Merae Alshahrani.

**Methodology:** Ahmed Abdullah Al Awadh, Yasir Mohammed Althayrayan, Majed Ahmed Althayrayan, Mohammed Merae Alshahrani.

**Project administration:** Ahmed Abdullah Al Awadh, Mohammed Merae Alshahrani.

**Resources:** Ahmed Abdullah Al Awadh, Mohammed Merae Alshahrani.

**Software:** Ahmed Abdullah Al Awadh, Osman A.E. Elnoubi, Mohammed Merae Alshahrani.

**Supervision:** Ahmed Abdullah Al Awadh, Mohammed Merae Alshahrani.

**Validation:** Ahmed Abdullah Al Awadh, Saadalnour Abusail Mustafa, Mohammed Merae Alshahrani.

**Visualization:** Ahmed Abdullah Al Awadh, Mohammed Merae Alshahrani.

**Writing – original draft:** Ahmed Abdullah Al Awadh, Elhashimi Eltayb Hassan, Omer Mohamed Shoaib, Saadalnour Abusail Mustafa, Mohammed Merae Alshahrani.

**Writing – review & editing:** Ahmed Abdullah Al Awadh, Elhashimi Eltayb Hassan, Omer Mohamed Shoaib, Osman A.E. Elnoubi, Saadalnour Abusail Mustafa, Mohammed Merae Alshahrani.

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
