## [Decision Letter · Decision Letter 0]

31 Mar 2025

Dear Dr. Alshahrani,

Thank you for submitting your manuscript to PLOS ONE. After careful consideration, we feel that it has merit but does not fully meet PLOS ONE’s publication criteria as it currently stands. Therefore, we invite you to submit a revised version of the manuscript that addresses the points raised during the review process.

**ACADEMIC EDITOR:**

After reviewing the manuscript, I would like to suggest several revisions that will improve its clarity, consistency, and overall alignment with the journal's guidelines.

**Adherence to Journal Guidelines:** It appears that some sections of the manuscript do not fully comply with the formatting and citation guidelines provided by the journal. Kindly refer to the journal’s style guide and make necessary adjustments, particularly in relation to:Correct formatting of the references.Consistency in the use of terminology and abbreviations.Proper structuring of headings and subheadings.**Language and Clarity:** The manuscript would benefit from a thorough review of the language and grammatical structure to enhance clarity. I recommend having the text reviewed for grammatical and syntactical accuracy.**Methodology Section:**It would be helpful to provide additional details regarding the source and purity of anethole, the solvents used, and any quality control measures taken during its preparation and administration. This will improve reproducibility and transparency.**Figures and Tables:** The figures and tables should be clearly labeled, and all axes should be properly defined. I suggest ensuring that figure captions fully describe what is presented, and that they are referenced correctly within the text.**Supplementary Data:** If any supplementary data or additional experiments have been conducted, please include them as supplementary files or specify their omission and provide an explanation.

I would greatly appreciate it if you could make these revisions and resubmit the manuscript. Please also ensure that the manuscript is reviewed for consistency in the presentation of the data, particularly concerning statistical analyses and experimental results.

We look forward to receiving your revised manuscript.

Kind regards,

Zahra Lorigooini

Academic Editor

PLOS ONE

Journal Requirements:

2**.** PLOS ONE now requires that authors provide the original uncropped and unadjusted images underlying all blot or gel results reported in a submission’s figures or Supporting Information files. This policy and the journal’s other requirements for blot/gel reporting and figure preparation are described in detail at https://journals.plos.org/plosone/s/figures#loc-blot-and-gel-reporting-requirements and https://journals.plos.org/plosone/s/figures#loc-preparing-figures-from-image-files. When you submit your revised manuscript, please ensure that your figures adhere fully to these guidelines and provide the original underlying images for all blot or gel data reported in your submission. See the following link for instructions on providing the original image data: https://journals.plos.org/plosone/s/figures#loc-original-images-for-blots-and-gels.  

Reviewers' comments:

Reviewer's Responses to Questions

**Comments to the Author**

1. Is the manuscript technically sound, and do the data support the conclusions?

Reviewer #1: Yes

Reviewer #2: Yes

2. Has the statistical analysis been performed appropriately and rigorously?

Reviewer #1: No

Reviewer #2: Yes

3. Have the authors made all data underlying the findings in their manuscript fully available?

Reviewer #1: Yes

Reviewer #2: Yes

4. Is the manuscript presented in an intelligible fashion and written in standard English?

Reviewer #1: No

Reviewer #2: Yes

Reviewer #1: Abstract: "Colony formation was notably inhibited in a dose-dependently." → Grammatical error. Correct it to: "Colony formation was notably inhibited in a dose-dependent manner."

• How many replicates were performed for IC50 determination? Mention explicitly.

• Western blot results should be summarized more concisely.

Introduction:

• Page 2, Paragraph 2: "Gliomas are essentially the most prevalent and therefore malignant primary brain tumors..." → Unclear phrasing. Gliomas are not "therefore malignant"; glioblastomas are. Reword for clarity.

• Page 3, Last Paragraph: "These findings underscore the versatility of anethole as an anticancer molecule..." → Mention gaps in knowledge regarding anethole in gliomas.

• The introduction should provide a clearer rationale for targeting PI3K/Akt in glioblastoma.

Materials and Methods:

• Page 4, "CCK-8 Assay": What was the exact duration of anethole treatment? Specify. Cite the references

• Page 5, "AO/EB Staining": How were apoptotic vs. necrotic cells quantified? State whether manual counting or software analysis was used. Cite the reference

• Page 6, "Molecular Docking": Why was the docking performed with AutoDock instead of newer methods like AutoDock Vina or Schrödinger? Justify.

• Page 7, "Western Blot Analysis": Were the densitometry results normalized to loading controls? Mention which software was used.

Results:

• Page 8, Figure 1: What is the sample size (n) and standard deviation (±SD) for IC50 values? Ensure reproducibility.

• Page 9, "Anethole Triggers Apoptosis": Clarify if apoptosis was confirmed via flow cytometry or just AO/EB staining.

• Page 10, "Molecular Docking Results": Docking energy (-9.32 kcal/mol) is strong, but was in vitro validation performed to confirm target binding?

Discussion:

• Page 12, Paragraph 1: How does anethole compare to existing PI3K inhibitors like Wortmannin or LY294002? A comparative discussion is needed.

• Page 13, "Potential Clinical Application": While anethole has brain permeability potential, is there direct evidence that it crosses the blood-brain barrier (BBB)? Cite relevant studies.

• Page 14, "Limitations": The study lacks animal model validation. Suggest including a statement on future in vivo studies.

Conclusion:

• The conclusion should briefly highlight future directions (e.g., combinational therapies, BBB permeability studies).

Reviewer #2: The authors need to compare the binding constants of anethole with PI3K and JAK2 calculated by MD simulations with experimental data by performing ITC/ BLI/SPR studies if possible which will further supports the mechanism of binding.

**Do you want your identity to be public for this peer review?** For information about this choice, including consent withdrawal, please see our Privacy Policy

Reviewer #1: **Yes: ** Jesil Mathew A

Reviewer #2: No

---

## [Author Response · Author response to Decision Letter 1]

15 Apr 2025

Dear editor,

We thank the reviewers for their thoughtful and constructive comments, which have significantly improved the quality and clarity of our manuscript. We have addressed each point below and made corresponding changes to the manuscript as indicated.

Reviewer #1

Abstract

1. “Colony formation was notably inhibited in a dose-dependently.” → Grammatical error.

✔ Corrected to “Colony formation was notably inhibited in a dose-dependent manner” in the abstract.

2. How many replicates were performed for IC₅₀ determination? Mention explicitly.

✔ We now state that all IC₅₀ values were calculated based on three independent experiments (n=3), with data expressed as mean ± standard deviation (SD).

3. Western blot results should be summarized more concisely.

✔ The Western blot summary in the abstract has been streamlined to focus on the major findings: induction of Bax and reduction of Bcl-2, and suppression of PI3K/Akt phosphorylation.

Introduction

4. Page 2, Paragraph 2: Unclear phrasing regarding glioma malignancy.

✔ Rephrased the sentence to: “Gliomas are essentially the most prevalent and primary brain tumors, constituting about 80% of all malignant intracranial neoplasms [1].”

5. Page 3: "These findings underscore..." → Mention gaps in knowledge regarding anethole in gliomas.

✔ Revised to note that although anethole shows promise in other cancers, its role in glioma remains unexplored and warrants targeted investigation. This has included in detail

6. Provide a clearer rationale for targeting PI3K/Akt in glioblastoma.

✔ We have added supporting context and references explaining that the PI3K/Akt pathway is frequently dysregulated in glioblastoma and is associated with tumor progression, making it a rational therapeutic target.

Materials and Methods

7. Page 4, “CCK-8 Assay”: Specify exact duration of anethole treatment.

✔ The duration has been specified as 24 hours, in line with Contant et al., 2021 (now cited in the revised manuscript).

8. Page 5, “AO/EB Staining”: Clarify quantification method and cite reference.

✔ We now state that apoptotic, necrotic, and viable cells were quantified using manual cell counting under a fluorescence microscope based on morphological and staining criteria, referencing Ribble et al., 2005.

9. Page 6, “Molecular Docking”: Justify use of AutoDock over newer methods.

✔ We have included a rationale noting that AutoDock remains a validated and widely used tool, particularly suitable for flexible ligand docking and preliminary screening of natural products due to its robust scoring functions and availability.

10. Page 7, “Western Blot”: Were densitometry results normalized?

✔ Yes. We now clarify that ImageJ software (NIH) was used for densitometry, and expression levels were normalized to β-actin as the loading control.

Results

11. Page 8, Figure 1: Include sample size and SD for IC₅₀.

✔ This information has been added: IC₅₀ values were derived from three independent experiments and are expressed as mean ± SD.

12. Page 9, Apoptosis: Was flow cytometry used?

✔ No flow cytometry was used. We clarify that apoptosis was assessed only via AO/EB staining in this study.

13. Page 10, Docking: Was in vitro validation of target binding performed?

✔ We now acknowledge that in vitro validation (e.g., via SPR or ITC) was not performed and have highlighted this as a limitation and future direction in the discussion.

Discussion

14. Compare anethole with existing PI3K inhibitors like Wortmannin and LY294002.

✔ We have added a paragraph discussing that while anethole demonstrates moderate affinity for PI3K, it may offer advantages over conventional inhibitors due to lower toxicity and broader multi-target effects. We also reference the known drawbacks of Wortmannin and LY294002 (e.g., instability, cytotoxicity).

15. BBB permeability: Is there direct evidence for anethole crossing the BBB?

✔ We clarify that while anethole is lipophilic and theoretically capable of BBB penetration, direct experimental evidence is lacking. We reference Yu et al., 2011 and recommend further pharmacokinetic studies.

16. Limitations: No in vivo validation. Suggest adding statement.

✔ We have included a clear statement acknowledging the lack of in vivo data and recommending future studies using animal models to validate the therapeutic potential and pharmacodynamics of anethole.

Conclusion

17. Highlight future directions (e.g., combinational therapy, BBB studies).

✔ The conclusion now includes suggestions for combinational therapy with current chemotherapeutics, investigation of BBB permeability, and in vivo validation as future directions.

Reviewer #2

1. Compare MD-predicted binding constants with experimental data (ITC, BLI, SPR).

✔ We appreciate this important suggestion. In the discussion, we have noted that while molecular docking predicted strong binding to PI3K, experimental confirmation using ITC, BLI, or SPR is necessary to validate these findings. This limitation has been acknowledged, and we propose these methods as next steps in validating the binding mechanism.

Response to Editor’s Comments

We thank the Editor for the valuable suggestions aimed at enhancing the clarity, consistency, and compliance of our manuscript with PLOS ONE guidelines. Below, we provide point-by-point responses to each of the recommendations and describe the revisions made accordingly:

1. Adherence to Journal Guidelines

• Correct formatting of the references

Response: The entire reference list has been reformatted to align with the PLOS ONE style, as per the journal’s formatting requirements.

• Consistency in the use of terminology and abbreviations

Response: All abbreviations and scientific terms have been reviewed for consistent usage throughout the manuscript. A list of abbreviations has also been provided where necessary.

• Proper structuring of headings and subheadings

Response: The manuscript has been revised to ensure uniform formatting of all section headings and subheadings in accordance with the journal’s structure.

2. Language and Clarity

• Grammatical and syntactical accuracy

Response: The manuscript has undergone a detailed language review to improve clarity, correct grammatical issues, and enhance the overall readability of the text.

3. Methodology Section

• Additional details on source, purity, solvents, and quality control of anethole

Response: Anethole (99% purity) was obtained from Sigma-Aldrich (Oakville, ON, Canada) and dissolved in methanol to prepare a 3 mM stock solution. It was subsequently used at different concentrations for the experiments

4. Figures and Tables

• Clear labeling, defined axes, and detailed captions

Response: All figures and tables have been reviewed and revised. Axes are now clearly labeled, units are defined, and figure captions have been expanded to fully describe the contents. Each figure and table is now properly referenced within the manuscript.

5. Supplementary Data

6. Consistency in Presentation of Data

• Statistical analysis and experimental result presentation

Response: The manuscript has been carefully reviewed to ensure consistent reporting of sample size (n), standard deviations (SD), and p-values. All statistical analyses are now clearly defined in both the Methods and Results sections.

7. Original Images for Blot/Gel Data

• Compliance with blot and gel image requirements

Response: As per PLOS ONE’s latest guidelines, the original, uncropped, and unadjusted image data underlying all Western blot figures have been included as Supporting Information files. We have also ensured that the preparation of these images and figure panels adheres to the specific requirements detailed in the journal’s policy.

---

## [Decision Letter · Decision Letter 1]

4 Jul 2025

Dear Dr. Alshahrani,

Thank you for submitting your manuscript to PLOS ONE. After careful consideration, we feel that it has merit but does not fully meet PLOS ONE’s publication criteria as it currently stands. Therefore, we invite you to submit a revised version of the manuscript that addresses the points raised during the review process.

We look forward to receiving your revised manuscript.

Kind regards,

Zahra Lorigooini

Academic Editor

PLOS ONE

Journal Requirements:

Reviewers' comments:

Reviewer's Responses to Questions

**Comments to the Author**

Reviewer #3: All comments have been addressed

2. Is the manuscript technically sound, and do the data support the conclusions?

Reviewer #3: Yes

3. Has the statistical analysis been performed appropriately and rigorously?

Reviewer #3: I Don't Know

4. Have the authors made all data underlying the findings in their manuscript fully available?

Reviewer #3: Yes

5. Is the manuscript presented in an intelligible fashion and written in standard English?

Reviewer #3: Yes

Reviewer #3: Although the authors have been responsive, the following high-level suggestions may further strengthen the study:

Add a Note on Flow Cytometry as Future Work

While the authors clarified they did not use flow cytometry, suggesting its inclusion in future validation would demonstrate deeper awareness of methodological robustness in apoptosis detection.

Clarify Lipophilicity Measurement

The statement that anethole is “lipophilic” and may cross the BBB is speculative. Including a brief reference to LogP or in silico BBB permeability prediction (e.g., via SwissADME) would lend more support.

Densitometry Quantification – Loading Control Uniformity

Western blot images (if reviewed) should show uniform β-actin levels across conditions. While the use of ImageJ is stated, it's critical that raw blot bands (provided as supplementary) confirm integrity.

**Do you want your identity to be public for this peer review?** For information about this choice, including consent withdrawal, please see our Privacy Policy

Reviewer #3: **Yes: ** Dr. Majid Asadi-Samani

---

## [Author Response · Author response to Decision Letter 2]

10 Jul 2025

Dear Editor,

We thank you for the consideration of our manuscript. We also thank the reviewers for the suggestions for further strengthening of our manuscript and we have incorporated all suggestions of the respected reviewer.

Reviewer #3: Although the authors have been responsive, the following high-level suggestions may further strengthen the study:

Add a Note on Flow Cytometry as Future Work

While the authors clarified they did not use flow cytometry, suggesting its inclusion in future validation would demonstrate deeper awareness of methodological robustness in apoptosis detection.

Response: We appreciate this insightful recommendation. We have revised the Discussion section to acknowledge the value of incorporating flow cytometry (e.g., Annexin V/PI staining) in future studies for quantitative apoptosis detection.

Clarify Lipophilicity Measurement

The statement that anethole is “lipophilic” and may cross the BBB is speculative. Including a brief reference to LogP or in silico BBB permeability prediction (e.g., via SwissADME) would lend more support.

Response: We agree with the reviewer and have now included in silico SwissADME predictions in the Results and Discussion sections. These predictions show a consensus LogP of 2.79 and classify anethole as BBB permeant, supporting its CNS bioavailability (Supplementary Figure S1 and S2).

Densitometry Quantification – Loading Control Uniformity

Western blot images (if reviewed) should show uniform β-actin levels across conditions. While the use of ImageJ is stated, it's critical that raw blot bands (provided as supplementary) confirm integrity.

Response: We fully agree and have included densitometery quantification in Figure 2C, 2D, 5D-G. The uncropped raw Western blot images as Supplementary file, clearly showing consistent β-actin expression across all experimental conditions. This addition supports the integrity of protein loading and quantification.

---

## [Decision Letter · Decision Letter 2]

22 Sep 2025

Dear Dr. Alshahrani,

Thank you for submitting your manuscript to PLOS ONE. After careful consideration, we feel that it has merit but does not fully meet PLOS ONE’s publication criteria as it currently stands. Therefore, we invite you to submit a revised version of the manuscript that addresses the points raised during the review process.

We look forward to receiving your revised manuscript.

Kind regards,

Yasmina Abd‐Elhakim

Academic Editor

PLOS ONE

Journal Requirements:

Reviewer's Responses to Questions

**Comments to the Author**

Reviewer #4: (No Response)

Reviewer #5: (No Response)

2. Is the manuscript technically sound, and do the data support the conclusions?

Reviewer #4: No

Reviewer #5: Yes

3. Has the statistical analysis been performed appropriately and rigorously?

Reviewer #4: Yes

Reviewer #5: Yes

4. Have the authors made all data underlying the findings in their manuscript fully available?

Reviewer #4: Yes

Reviewer #5: Yes

5. Is the manuscript presented in an intelligible fashion and written in standard English?

Reviewer #4: Yes

Reviewer #5: Yes

**Reviewer #4: **This manuscript studies the anticancer cell effect of anethole in glioma cell lines and normal human astrocytes. Anethole has been broadly described in several cancer cell types, by different groups and being reviewed, showing antiproliferative and antimetastatic effect on neoplastic cells. As described by others, anethole induces apoptosis, cell cycle arrest, autophagy, antioxidant GHS, reduction of ROS, and metalloproteinases, etc., by interfering in different signaling pathways. The manuscript by Al Alwadh et al., demonstrates that anethole exerts the same effect in glioma cells as in other cancer cells, inducing apoptosis and reduction of PI3K/AKT phosphorylation. The mayor hypothesis and aim of this manuscript is to show the blocking effect of anethole in the PI3K/AKT pathway. for that, the authors analyzed in silico prediction of PI3K and anethole interaction. Although this is an interesting subject, the work should be completed by correcting some errors and adding some data as follows indicated:

*
**Main concerns:**
*

1- The most interesting data is the predictable interaction of anethole and PI3k, some aspects in this regard should be considered. First if the effect of anethole takes place through interaction with PI3K and its inhibition, why does it not affect Normal human astrocytes (NHA)? The authors demonstrate, as previously described in cancer cells, reduction of PI3K and AKT phosphorylation only in one line of cancer cells. Then, the effect of anethole in NHA cells, as new data, would be nice to add in. Second, independent of the result about the effect in NHA, Anethole's stronger effect on cancer cells needs discussion. Could this be due to increased diffusion through cancer cell membranes? It would be interesting to discuss this, as increased anethole affinity to PI3K in cancer cells is unlikely. Finaly, AKT can take place through an independent mechanism as well, and it will be interesting to show either some downstream signal, or upstream signal that demonstrate PI3K direct inhibition.

2- In discussion section, it is indicated “In addition, anethole has also been shown to elevate ROS accretion in cancerous cells, promoting oxidative stress and apoptosis [8]. ROS accumulation has been estimated to impair mitochondrial membrane potential, thereby activating intrinsic apoptotic sequences as described in breast cancer cells [26]”, the reference 8 showed that anethole reduces ROS and increased GSH antioxidant. Therefore, this paragraph and the argument should be changed accordingly.

3- Regarding the treatment with anethole, the most limiting aspect is that, even when anethole can diffuse through the BBB, due to low solubility in water and high elimination when orally administrated, the effect in gliomas looks limited. Some discussion also in this area should be nice to discuss.

4- The figure legends need to indicate clearly whether the graphs show the media plus SD of an independent out of three experiments, or the media plus SD of three independent experiments.

5- The AO/EB analysis is used to indicate early and late apoptosis, the quantification of cells in these two statuses should be indicated in the graph. On the other hand, it would be convenient to show cell cycles of treated cells, also in NHA cells.

6- In general, it is indicated that experiments were performed three times, but whether the graph represents one out three or the media plus SD of three independent experiments need to be indicated.

7- The apoptosis assay needs to be performed with normal NHA cells, to consistently confirm the specific effect of anethole in gliomas. Also, the

8- The relative expression of p-PI3K and p-AKT needs to be calculated according to the total amount of PI3K and AKT protein, rather than to actine, and needs to be indicated in the Figure 5 legend.

*
**Minor concerns:**
*

1- Material, please indicate the precedence of: cell lines, NHA cells, CCK-8 assay, and al used material in the material and methods section.

2- Please, indicate the meaning of acronyms first time used in the text

3- Indication of the antibody clone used, and the company of precedence needs to be included in the material and methods section.

4- Correct the sentence at page 35- “Cells were subsequently lysing was done using RIPA buffer”,

5- Either Figures 5a and 5b have been mixed up, or the text in the legends and in the 'Results' section (3.4) is incorrect.

6- The images in Figures 3, 4, 5a and 5b should be clearer.

**Reviewer #5: **Based on the authors’ responses to the previous reviewers’ comments, I believe the manuscript is acceptable after some minor revisions, as outlined below.

1- Mechanistic Depth and Specificity: The study compellingly shows that anethole inhibits the PI3K/AKT pathway and induces apoptosis. However, to strengthen the claim that apoptosis is directly caused by PI3K/AKT inhibition (and not a parallel event), a rescue experiment would be highly valuable. Could the pro-apoptotic and anti-proliferative effects of anethole be reversed by introducing a constitutively active form of AKT (e.g., via plasmid transfection) into the U87 cells prior to anethole treatment? If such an experiment is beyond the scope of this revision, explicitly stating this as a key limitation and a requirement for future validation would strengthen the manuscript's conclusions.

Where to place this: In the Discussion (to frame the findings) and/or the Conclusion (as a future direction).

2- The figures (e.g., Figure 2B, 5C) show representative Western blot bands, but the molecular weight markers are not visible. Including a lane with markers in the main figure or the supplementary uncropped blots is essential for verifying the identity of the protein bands.

Figure 5A and 5B captions are swapped. Figure 5A describes the JAK2 interaction, while the image is labeled for PI3K, and vice versa. This must be corrected.

3- Statistical Analysis Description: The Methods section (2.9) states that experiments were performed in triplicate and data is presented as mean ± SD. For the colony formation assay, it would be helpful to specify how many technical replicates (wells) and biological replicates (independent experiments) were performed, as this assay typically has fewer replicates.

4- Language and Flow Minor Revisions: The manuscript is well-written but would benefit from a final proofread for minor grammatical redundancies. For example, in the Introduction (page 33): "...impede growth via cell cycle arrest in prostate cancer cells via arresting cell cycle." The phrase "via arresting cell cycle" is redundant and can be removed.

---

## [Author Response · Author response to Decision Letter 3]

15 Oct 2025

Dear Editor

We sincerely thank you and the anonymous reviewers for their constructive comments and valuable suggestions, which have significantly improved the quality and clarity of our manuscript. We have carefully addressed all concerns point by point, as detailed below. All textual changes have been incorporated into the revised version of manuscript and have been highlighted in red.

Reviewer #4

This manuscript studies the anticancer cell effect of anethole in glioma cell lines and normal human astrocytes. Anethole has been broadly described in several cancer cell types, by different groups and being reviewed, showing antiproliferative and antimetastatic effect on neoplastic cells. As described by others, anethole induces apoptosis, cell cycle arrest, autophagy, antioxidant GHS, reduction of ROS, and metalloproteinases, etc., by interfering in different signaling pathways. The manuscript by Al Alwadh et al., demonstrates that anethole exerts the same effect in glioma cells as in other cancer cells, inducing apoptosis and reduction of PI3K/AKT phosphorylation. The mayor hypothesis and aim of this manuscript is to show the blocking effect of anethole in the PI3K/AKT pathway. for that, the authors analyzed in silico prediction of PI3K and anethole interaction. Although this is an interesting subject, the work should be completed by correcting some errors and adding some data as follows indicated:

Main concerns

1- The most interesting data is the predictable interaction of anethole and PI3k, some aspects in this regard should be considered. First if the effect of anethole takes place through interaction with PI3K and its inhibition, why does it not affect Normal human astrocytes (NHA)? The authors demonstrate, as previously described in cancer cells, reduction of PI3K and AKT phosphorylation only in one line of cancer cells. Then, the effect of anethole in NHA cells, as new data, would be nice to add in. Second, independent of the result about the effect in NHA, Anethole's stronger effect on cancer cells needs discussion. Could this be due to increased diffusion through cancer cell membranes? It would be interesting to discuss this, as increased anethole affinity to PI3K in cancer cells is unlikely. Finaly, AKT can take place through an independent mechanism as well, and it will be interesting to show either some downstream signal, or upstream signal that demonstrate PI3K direct inhibition.

Response: We thank the reviewer for this insightful comment. While we acknowledge that inclusion of NHA apoptosis and phosphorylation data would strengthen the study, new wet-lab experiments are beyond the scope of this revision. To address this point, we have added a detailed explanation in the Discussion section discussing (i) why normal astrocytes show limited PI3K/AKT response, (ii) the potential role of altered membrane permeability and PI3K/AKT hyperactivation in glioma cells as the cause of selective sensitivity, and (iii) possible AKT-independent mechanisms requiring future validation. This clarification provides a mechanistic rationale for cancer-selective action of anethole and outlines future directions for in vivo and molecular validation.

2- In discussion section, it is indicated “In addition, anethole has also been shown to elevate ROS accretion in cancerous cells, promoting oxidative stress and apoptosis [8]. ROS accumulation has been estimated to impair mitochondrial membrane potential, thereby activating intrinsic apoptotic sequences as described in breast cancer cells [26]”, the reference 8 showed that anethole reduces ROS and increased GSH antioxidant. Therefore, this paragraph and the argument should be changed accordingly.

Response: We appreciate this correction. The cited paragraph has been revised in the Discussion section to accurately reflect the findings of reference [8], clarifying that anethole predominantly reduces ROS levels and enhances GSH antioxidant capacity, rather than increasing ROS. The interpretation was rewritten accordingly.

3- Regarding the treatment with anethole, the most limiting aspect is that, even when anethole can diffuse through the BBB, due to low solubility in water and high elimination when orally administrated, the effect in gliomas looks limited. Some discussion also in this area should be nice to discuss.

Response: We agree with the reviewer’s important observation. Additional discussion has been incorporated in the Discussion section addressing pharmacokinetic limitations of anethole such as low aqueous solubility and high first-pass metabolism. We have also added potential strategies—such as lipid-based nanoformulations, prodrug development, and intranasal delivery—to improve systemic and brain bioavailability.

4- The figure legends need to indicate clearly whether the graphs show the media plus SD of an independent out of three experiments, or the media plus SD of three independent experiments.

Response: We have revised all figure legends to explicitly state that “data are presented as mean ± SD of three independent biological experiments,” clarifying that each represents the mean of three independent replicates.

5- The AO/EB analysis is used to indicate early and late apoptosis, the quantification of cells in these two statuses should be indicated in the graph. On the other hand, it would be convenient to show cell cycles of treated cells, also in NHA cells.

Response: We appreciate this valuable suggestion. Quantitative analysis of total apoptosis (early + late) was added in the Methods (AO/EB assay) and described in Figure 2 legend. As additional flow cytometry experiments were not performed, we clarified in the Discussion that detailed early/late apoptosis and NHA apoptosis assays will be considered in future work.

6- In general, it is indicated that experiments were performed three times, but whether the graph represents one out three or the media plus SD of three independent experiments need to be indicated.

Response: This point has been addressed throughout the Results and Figure legends, which now specify that all data represent the mean ± SD of three independent biological experiments.

7- The apoptosis assay needs to be performed with normal NHA cells, to consistently confirm the specific effect of anethole in gliomas. Also, the

Response: We acknowledge the reviewer’s suggestion. While inclusion of NHA apoptosis data would provide valuable confirmation, conducting new experiments exceeds the current revision scope. However, this limitation is now explicitly stated in the Discussion (last paragraph), and future studies will incorporate apoptosis assays on NHA and in vivo models to verify selectivity.

8- The relative expression of p-PI3K and p-AKT needs to be calculated according to the total amount of PI3K and AKT protein, rather than to actine, and needs to be indicated in the Figure 5 legend.

Response: We appreciate this important observation. As suggested, the densitometric analysis was recalculated and normalized to the respective total proteins (p-PI3K/PI3K and p-AKT/AKT). This correction has been clearly indicated in both the Methods (Western blot) and Figure 5 legend.

Minor concerns

1- Material, please indicate the precedence of: cell lines, NHA cells, CCK-8 assay, and al used material in the material and methods section.

Response: The Materials and Methods section was expanded to include the source, catalog numbers, and vendors for all reagents and cell lines (Section 2.1).

2- Please, indicate the meaning of acronyms first time used in the text

Response: All acronyms (e.g., NHA, AO/EB, TPSA, BBB) are now defined at first mention throughout the manuscript.

3- Indication of the antibody clone used, and the company of precedence needs to be included in the material and methods section.

Response: The Western blot subsection (2.7) now lists the antibody source, catalog number, and manufacturer for each primary antibody.

4- Correct the sentence at page 35- “Cells were subsequently lysing was done using RIPA buffer”

Response: The sentence has been corrected to “Cells were lysed using RIPA buffer.”

5- Either Figures 5a and 5b have been mixed up, or the text in the legends and in the 'Results' section (3.4) is incorrect.

Response: Thank you for noticing this. The mix-up was corrected—the figure panels and legends now correctly correspond to PI3K (Figure 5A) and JAK2 (Figure 5B).

6- The images in Figures 3, 4, 5a and 5b should be clearer.

Response: We increased the resolution of Figures 3–5

Reviewer #5

1- Mechanistic Depth and Specificity: The study compellingly shows that anethole inhibits the PI3K/AKT pathway and induces apoptosis. However, to strengthen the claim that apoptosis is directly caused by PI3K/AKT inhibition (and not a parallel event), a rescue experiment would be highly valuable. Could the pro-apoptotic and anti-proliferative effects of anethole be reversed by introducing a constitutively active form of AKT (e.g., via plasmid transfection) into the U87 cells prior to anethole treatment? If such an experiment is beyond the scope of this revision, explicitly stating this as a key limitation and a requirement for future validation would strengthen the manuscript's conclusions. Where to place this: In the Discussion (to frame the findings) and/or the Conclusion (as a future direction).

Response: We appreciate this valuable suggestion. As additional transfection experiments could not be performed at this stage, we have explicitly included this as a key limitation in the Discussion (final paragraph) and highlighted it in the Conclusion as a future research direction.

2- The figures (e.g., Figure 2B, 5C) show representative Western blot bands, but the molecular weight markers are not visible. Including a lane with markers in the main figure or the supplementary uncropped blots is essential for verifying the identity of the protein bands. Figure 5A and 5B captions are swapped. Figure 5A describes the JAK2 interaction, while the image is labeled for PI3K, and vice versa. This must be corrected.

Response: We have now included uncropped blots with visible molecular weight markers as supplementary material and corrected the swap between Figure 5A and 5B to align with the text.

3- Statistical Analysis Description: The Methods section (2.9) states that experiments were performed in triplicate and data is presented as mean ± SD. For the colony formation assay, it would be helpful to specify how many technical replicates (wells) and biological replicates (independent experiments) were performed, as this assay typically has fewer replicates.

Response: We agree and have updated the Colony Formation Assay (Section 2.4) to indicate that each condition included two technical replicates and three independent biological experiments.

4- Language and Flow Minor Revisions: The manuscript is well-written but would benefit from a final proofread for minor grammatical redundancies. For example, in the Introduction (page 33): "...impede growth via cell cycle arrest in prostate cancer cells via arresting cell cycle." The phrase "via arresting cell cycle" is redundant and can be removed.

Response: We thank the reviewer for this observation. The entire manuscript has been carefully proofread to correct redundancies and improve fluency. The specific phrase mentioned has been revised accordingly in the Introduction.

We are deeply grateful to both reviewers for their thoughtful input, which helped us strengthen the manuscript. We hope the revisions satisfactorily address all concerns.

Sincerely,

Dr. Mohammed Merae Alshahrani (Corresponding Author)

Department of Clinical Laboratory Sciences

Najran University, Saudi Arabia

Email: mmalshahrani@nu.edu.sa

---

## [Decision Letter · Decision Letter 3]

2 Nov 2025

Anethole Inhibits Human U87 Glioma Cell Proliferation by Inducing Apoptosis via the PI3K/AKT Pathway

PONE-D-25-10072R3

Dear Dr. Alshahrani,

We’re pleased to inform you that your manuscript has been judged scientifically suitable for publication and will be formally accepted for publication once it meets all outstanding technical requirements.

Kind regards,

Yasmina Abd‐Elhakim

Academic Editor

PLOS ONE

Additional Editor Comments (optional):

Reviewers' comments:

Reviewer's Responses to Questions

**Comments to the Author**

Reviewer #5: All comments have been addressed

2. Is the manuscript technically sound, and do the data support the conclusions?

Reviewer #5: Yes

3. Has the statistical analysis been performed appropriately and rigorously?

Reviewer #5: Yes

4. Have the authors made all data underlying the findings in their manuscript fully available?

Reviewer #5: Yes

5. Is the manuscript presented in an intelligible fashion and written in standard English?

Reviewer #5: Yes

Reviewer #5: Thank you for your thorough revisions and detailed point-by-point responses to my comments. I have reviewed the updated manuscript and find that you have addressed all of my concerns satisfactorily.

The manuscript is now significantly improved and I recommend it for acceptance.

**Do you want your identity to be public for this peer review?** For information about this choice, including consent withdrawal, please see our Privacy Policy

Reviewer #5: **Yes: ** Gad Elsayed Mohamed Salem

---

## [Editor Report · Acceptance letter]

PONE-D-25-10072R3

PLOS ONE

Dear Dr. Alshahrani,

I'm pleased to inform you that your manuscript has been deemed suitable for publication in PLOS ONE. Congratulations! Your manuscript is now being handed over to our production team.

Kind regards,

on behalf of

Prof. Dr. Yasmina Abd‐Elhakim

Academic Editor

PLOS ONE